# Genetic and Pharmacological Inhibition of Autophagy Increases the Monoubiquitination of Non-Photosynthetic Phospho*enol*pyruvate Carboxylase

**DOI:** 10.3390/plants10010012

**Published:** 2020-12-23

**Authors:** Guillermo Baena, Ana B. Feria, Luis Hernández-Huertas, Jacinto Gandullo, Cristina Echevarría, José A. Monreal, Sofía García-Mauriño

**Affiliations:** Departamento de Biología Vegetal y Ecología, Facultad de Biología, Universidad de Sevilla, Avenida Reina Mercedes nº 6, 41012 Seville, Spain; gbaena@us.es (G.B.); anabelen@us.es (A.B.F.); luishernanhuertas@gmail.com (L.H.-H.); jacintogt@us.es (J.G.); echeva@us.es (C.E.); monreal@us.es (J.A.M.)

**Keywords:** autophagy, *Arabidopsis thaliana*, *Nicotiana benthamiana*, phospho*enol*pyruvate carboxylase, ubiquitin

## Abstract

Phospho*enol*pyruvate carboxylase (PEPC) is an enzyme with key roles in carbon and nitrogen metabolisms. The mechanisms that control enzyme stability and turnover are not well known. This paper investigates the degradation of PEPC via selective autophagy, including the role of the monoubiquitination of the enzyme in this process. In Arabidopsis, the genetic inhibition of autophagy increases the amount of monoubiquitinated PEPC in the *atg2*, *atg5*, and *atg18a* lines. The same is observed in *nbr1*, which is deficient in a protein that recruits monoubiquitinated substrates for selective autophagy. In cultured tobacco cells, the chemical inhibition of the degradation of autophagic substrates increases the quantity of PEPC proteins. When the formation of the autophagosome is blocked with 3-methyladenine (3-MA), monoubiquitinated PEPC accumulates as a result. Finally, pull-down experiments with a truncated version of NBR1 demonstrate the recovery of intact and/or fragmented PEPC in Arabidopsis leaves and roots, as well as cultured tobacco cells. Taken together, the results show that a fraction of PEPC is cleaved via selective autophagy and that the monoubiquitination of the enzyme has a role in its recruitment towards this pathway. Although autophagy seems to be a minor pathway, the results presented here increase the knowledge about the role of monoubiquitination and the regulation of PEPC degradation.

## 1. Introduction

Phospho*enol*pyruvate carboxylase (PEPC; EC 4.1.1.31) is a key enzyme in the metabolism of carbon (C) and nitrogen (N), with central roles in photosynthesis, respiration, amino acid synthesis, and the development and germination of seeds [1,2]. This enzyme catalyzes the addition of bicarbonate to PEP to form the four-carbon compound oxaloacetate, which is reduced to malate by malate dehydrogenase (MDH). PEPC is mostly acknowledged by its role in C_4_ and Crassulacean Acid Metabolism (CAM) photosynthesis [1,3], though it also has key functions in C_3_ plants and C_3_ tissues, such as in seeds, fruits, roots, stomata, legume nodules, and others [2].

The PEPC gene family (*PPC* genes) includes plant-type PEPC (PTPC) and bacterial-type PEPC (BTPC) genes [4]. All PTPCs have a conserved N-terminal seryl residue that is phosphorylated by PEPC kinases (PEPCks). This residue is absent in BTPC [5]. In the model plant *Arabidopsis thaliana*, the *PPC* gene family consists of three PTPCs (*PPC1-3*) and one BTPC (*PPC4*) [6]. *PPC2* transcripts are found in all organs, suggesting that it is a housekeeping gene. The *PPC3* gene is expressed in roots and *PPC1* is expressed in both roots and flowers, as with *PPC4* [7]. The predicted PEPC proteins from a *Nicotiana benthamiana* draft genome [8] have been named on the basis of their homology to *A. thaliana* PEPCs (see the results section for more information).

PEPC is subjected to different post-translational modifications (PTMs), such as phosphorylation, monoubiquitination, NO-related PTMs (S-nitrosylation, Tyr-nitration), and oxidative stress-associated PTMs (carbonylation) [9,10,11,12,13]. In addition, the acetylation of Arabidopsis PEPCs has recently been reported [14]. These PTMs regulate PEPC activity and the turnover of the protein [13], although the biological functions of some of them are not yet fully understood.

Protein phosphorylation is the most studied PTM of PEPC. This enzyme is phosphorylated at a conserved N-terminal serine residue by PEPC kinase (PEPCk). This PTM results in a positive effect for the functional and regulatory properties of PEPC, decreasing its sensitivity to feedback inhibition by L-malate and increasing its affinity for the allosteric activator glucose-6-phosphate and V_max_ [9]. The phosphorylation of photosynthetic isoenzymes occurs during the light period in C_4_ plants [15] and the dark period in CAM plants [3]. The relevance of this PTM is shown in a PEPCk-deficient *Kalanchoë fedtschenkoi* iRNA line that shows perturbations for the CAM photosynthesis, carbohydrate metabolism, and circadian rhythms [16].

Ubiquitination is a fundamental biochemical process which controls numerous aspects of protein functions such as degradation, protein–protein interaction, and subcellular localization [17]. In different plants and tissues, two immunoreactive PEPC bands are usually detected, and the upper is thought to be a monoubiquitinated form of the lower. In fact, treatment with deubiquitinating enzyme 2 (USP2 core) causes the disappearance of the upper band. The monoubiquitination of PEPC occurs at a conserved lysine residue in castor oil seeds [18], *Hakea prostata* seeds [19], and sorghum seeds [10,11]. In germinating castor oil seeds, the monoubiquitination of PEPC changes the kinetic properties of the enzyme, interfering with its ability to bind PEP and enhancing sensitivity for the majority of its metabolite effectors [18]. In most of the physiological contexts that have been investigated, phosphorylation and monoubiquitination have modulated PEPC activity in opposite ways [18,19,20], and these two PTMs seem to be mutually restrictive. Nevertheless, the sorghum seed PEPC can be phosphorylated and monoubiquitinated at the same time [10,11]. This suggests that monoubiquitination could play other unknown physiological roles. In addition, ammonium stress increases the monoubiquitination of the sorghum root PEPC [12], and the meaning of this finding is still undetermined.

Autophagy is a highly conserved process used for the bulk and selective cleavage of cellular components. The element that is going to be cleaved is encapsulated in double-membrane vesicles, termed autophagosomes, and later degraded inside the plant vacuole [21]. The process is regulated and executed by a conserved set of proteins called autophagy-related (ATG) proteins [22]. Autophagy displays a central role maintaining cellular homeostasis by removing damaged elements and thus preventing the effects of their accumulation. In addition, autophagy ensures a good management of resources by recycling the components of the cleaved elements. This is especially important under stress; in this situation, autophagy allows the preservation of key metabolic elements, meanwhile non-essential or altered elements are processed and the constituents of the latter are used for maintaining the former [23,24].

Autophagy plays an important role in the remobilization of nutrients, especially in suboptimal nutrient conditions [25]. The yield and quality of grain largely depends on the mobilization of N from the senescent leaves to the seeds [26] and autophagy is integral to this process. An increasing body of experimental data shows that impairing autophagy has a great negative impact on nitrogen use efficiency and crop yield [27]. In Arabidopsis *atg* mutants, nitrogen use efficiency (NUE) is significantly decreased [28]. Similarly, maize autophagy mutants cannot efficiently remobilize N from old leaves to seeds [29]. On the contrary, the overexpression of *ATG* genes enhances the level of autophagy and increases NUE, both in Arabidopsis [30] and rice [31].

While bulk autophagy randomly sequesters cytosolic content, selective autophagy constitutes a specific and highly controlled degradation pathway [32]. Selective autophagy requires cargo receptors which mediate selective cargo recruitment in response to diverse intra- and extra-cellular signals. Cargo receptor molecules link the cargo to the phagophore membrane via their simultaneous interaction with the cargo and ATG8 proteins on the membrane.

In contrast to the single ATG8 gene present in the genome of yeast, ATG8 generally exists as a multiprotein family in eukaryotes. Arabidopsis has nine ATG8 isoforms, annotated as ATG8a to ATG8i [33], meanwhile six isoforms have been reported in rice [34]. ATG8 provides a docking site for cargo receptors that contain short peptide motifs called ATG8-interacting motifs/LC3-interacting regions (AIMs/LIRs) [35]. Cargo receptors bind simultaneously to cargo and lipidated ATG8 or ATG8 family members. AIMs are consensus F/W/Y-X-X-L/I/V sequences that can be predicted and identified by bioinformatics approaches [36,37]. Recently, a new binding motif (UIM, ubiquitin interacting motif) has been discovered on ATG8 that binds receptors containing the UIM [38].

The protein NBR1 (neighbor of the BRCA1 gene) is a cargo receptor that contributes to the autophagic clearance of ubiquitinated substrates. Plant NBR1 binds ubiquitin through a C-terminal ubiquitin-associated (UBA) domain and interacts with homologs of ATG8 via an evolutionary conserved AIM motif [39]. Arabidopsis NBR1 interacts with six of the nine Arabidopsis ATG8 proteins, and it is degraded in the vacuole in an autophagy-dependent manner. A similar protein (Joka2) has been identified in tobacco [40].

This work investigates the degradation of PEPC via autophagy and the role of NBR1 and the monoubiquitination of PEPC on its recruitment for selective autophagy. Different experimental approximations are used, including genetic and pharmacological inhibitions of autophagy, as well as pull-down experiments. The results show that a fraction of PEPC is cleaved by selective autophagy, and that monoubiquitination contributes to direct a fraction of C_3_ PEPC in this pathway.

## 2. Results

### 2.1. Increased Monoubiquitinated PEPC in Arabidopsis Mutants and Defective Autophagy

Several experimental results obtained by other researchers have suggested a relationship between the monoubiquitination of PEPC and autophagy. First, in Arabidopsis plants expressing trehalose-6-P-synthase, higher levels of trehalose-6-P are accompanied by decreased amounts of monoubiquitinated PEPC [20]. Trehalose is the product of the dephosphorylation of trehalose-6-P. Trehalose has been shown to trigger autophagy in human and animal models [41,42] and exists in the desiccation-tolerant grass *Tripogon loliiformis* [43].

Monoubiquitinated PEPC (p110) shows lower mobility in acrylamide gels than non-monoubiquitinated PEPC (p107). This allows the identification of monoubiquitinated PEPC in gels and immunoblots as the upper band of immunoreactive PEPC protein. When trehalose was supplied to Arabidopsis plants, this decreased the monoubiquitination of PEPC in the leaves and roots (Figure 1). Although trehalose has not been demonstrated to trigger autophagy in Arabidopsis, this result prompted us to investigate the link between autophagy and monoubiquitination of PEPC.

The processing of PEPC via autophagy was explored in Arabidopsis SALK lines knockout for autophagy-related genes (ATG). The *atg18a* line was defective in ATG18a, which is a phosphatidylinositol 3-phosphate (PI3P) effector that interacts with ATG2 and is required for the formation of autophagosomes during nutrient stress and senescence [44]. Likewise, ATG2 and ATG5 are parts of the core machinery of autophagy [22]. NBR1, on the contrary, functions in the recruitment of selected proteins towards the specific autophagy pathway [45].

The amounts of PEPC proteins were measured in the leaves of *atg2* and *atg5* Arabidopsis SALK lines (Figure 2). These lines had more immunoreactive PEPC protein than the wild-type line (Col-0). As the increase was also more noticeable when the gels were charged by units of enzymatic activity (Figure 2a) than by µg of protein (Figure 2b), the results suggest that inactive PEPC is accumulated as a consequence of impaired autophagy.

Similar results were obtained with the leaves and roots of *nbr1* (Figure 3a) and *atg18a* lines (Figure 3b). In *nbr1* leaves, the accumulation of monoubiquitinated PEPC was especially evident (Figure 3a, left), represented by the upper band, which was revealed by anti-ubiquitin antibodies (Figure 3a, right). Increased p110/p107 ratios were detected to different extents in all the defective autophagy lines (Table 1).

The accumulation of monoubiquitinated PEPC was observed mainly in standard conditions. In plants subjected to stress, the upper band (Ub-PEPC) typically decreased, both in Col-0 and autophagy-deficient SALK lines. As an example, this could be seen for Col-0 and for *atg18a* under N starvation (Figure 3b). The results in *atg18a*, whose bulk autophagy was expected to be severely compromised, indicated that Ub-PEPC could be also processed by autophagy-independent mechanisms.

Taken together, the results for Arabidopsis indicate that at least a fraction of C_3_-type PEPC was degraded via selective autophagy. The monoubiquitination of the protein could be marking a fraction of the PEPC protein towards this pathway. Nevertheless, autophagy is not the only main pathway by which PEPC is degraded.

### 2.2. Chemical Inhibition of Autophagy in Cultured N. benthamiana Cells Increased the Amount of Monoubiquitinated PEPC

The results obtained via the genetic inhibition of autophagy (Arabidopsis mutants) were further studied in terms of the chemical inhibition of the degradation of autophagic substrates. The following experiments were performed with cultured *N. benthamiana* cells. Tobacco cells were cultured in the dark and organic C was supplied in the form of sucrose. When sucrose was absent, PEPC activity decreased and was nearly undetectable by 10 d. After 5 d without sucrose, PEPC represented about 30% of the activity with sucrose and it was fully recovered by the addition of sucrose, showing that the decreased PEPC activity was not due to death of the cells (Figure 4a).

If sucrose starvation triggers the autophagic degradation of PEPC, the addition of inhibitors in this process should preserve the protein. Three different inhibitors were used to test this hypothesis. First, 3-methyladenine (3-MA) was used, which impedes the formation of autophagosomes by inhibiting phosphatidylinositol-3-kinase (PI3K) [46]. Concanamycin A (ConcA) inhibits vacuolar-type ATPase, preventing vacuolar acidification and blocking the degradation of autophagic bodies inside vacuoles [47]. E64 inhibits Cys-proteases and stabilizes autophagic bodies inside vacuoles [48]. The presence of either of the inhibitors increased the amount of immunoreactive PEPC protein in sucrose-starved cells (Figure 4b). Interestingly, 3-MA noticeably increased the amount of mono-ubiquitinated PEPC, both in the absence or in the presence of sucrose (Figure 4b,c). These results indicate that PEPC is processed via autophagy, both in control conditions and under sucrose starvation, and that monoubiquitination could be a mark for recruitment to selective autophagy. The differences between 3-MA and the other two inhibitors suggest that ubiquitin is removed from PEPC inside the autophagosome before its inclusion in the vacuole. It has been reported that ubiquitin and other ubiquitin-like modifiers are recycled along degradation processes [49]. In the same line, the ubiquitin-like modifier PE-ATG8 is deconjugated by ATG4 and then recycled [50].

Phylogenetic and sequence analysis of predicted *N. benthamiana* PEPCs was performed with data from the new draft sequence of the *N. benthamiana* genome [8]. Six complete PEPC sequences were found in the database. In order to evaluate the homology and their phylogenetic relationships, the entire predicted amino acid sequences of the six PEPCs were compared with the protein sequences of PEPCs from *A. thaliana*. From this alignment, a phylogenetic tree and a phylogenetic distance matrix were constructed (Figure 5).

The BTPCs (AtPPC4 and NbPPC4) constituted a group separated from PTPCs (Figure 5a). The PTPC proteins were named on the basis of their homology and phylogenetic distance as compared to the *A. thaliana* PEPCs. The five putative protein-encoded PTPCs in *N. benthamiana* had a high level of homology between them as the amino acid sequences were highly conserved. One of the proteins presented more homology to AtPPC2 and comprised an independent cluster with AtPPC2 (Figure 5a). Relationships among the other four PTPCs were unresolved and, if so, they were weakly supported in both the BI and ML analyses. Three of them (NbPPC3.1-3) are very similar and showed more homology and less evolutionary distances with respect to AtPPC3, while one (NbPPC1) was more similar to AtPPC1 (Figure 5b).

The identities of the two bands were revealed by the anti-PEPC antibodies that exist in extracts of cultured tobacco cells and were confirmed by MALDI-TOF MS/MS. Peptide mass fingerprinting revealed that both the upper (p110) (Appendix A) and the lower (p100) fingerprints matched with PPC3 (Appendix A). Although the score for the p100 band was under the desired value, the identification was supported by the likeness of the MALDI fingerprints for the two bands (Appendix A).

The next step was to confirm that the p110 band corresponded to the monoubiquitination of p100. PEPC present in crude extracts from different sources was immunoprecipitated by incubation with anti-PEPC antibodies [51]. Anti-ubiquitin antibodies exclusively revealed the p110 band in the precipitate from *N. benthamiana* leaves and roots, as well as *A. thaliana* leaves (Figure 6a). Similar results were obtained with *N. benthamiana* cells (Figure 6b) when 3-MA was added to the growing medium (lane 3). In the absence of 3-MA, the amount of monoubiquitinated PEPC was barely noticeable (lane 2). This result demonstrates that the upper band effectively corresponds to an ubiquitinated form of PEPC, and that the amount of monoubiquitinated PEPC was increased when the formation of autophagosomes was blocked.

### 2.3. Pull-Down with GST-NBR1 and GFP-ATG8a

Several experimental approaches were used to investigate a possible degradation of PEPC via recruitment by NBR1, which interacts with ubiquitinated proteins towards selective autophagy. First, experiments were carried out using truncated SbNBR1 containing UBA1, UBA2, and AIM, which was purified and subsequently used alongside glutathione agarose resin. This resin binds to glutathione *S*-transferase (GST) fusion proteins. Experiments were performed with Arabidopsis leaves and roots and *N. benthamiana* cultured cells.

Pull-down experiments were performed with leaves (Figure 7a) and roots (Figure 7b) of Arabidopsis lines Col-0, *atg2-2*, and *atg5-1*. Anti-PEPC antibodies revealed a 65 kDa peptide that interacted with GST-NBR1 in leaves and roots extracts from all the lines. In addition, a 100 kDa anti-PEPC immunoreactive peptide was observed in the leaves of the *atg5-1* line. The amounts of co-immunoprecipitated PEPCs were slightly higher in the autophagy-deficient lines.

The pull-down experiments with truncated GST-NBR1 in *N. benthamiana* cells extracts obtained different results in different conditions. Experiments were conducted with the control, without sucrose and/or in presence of the inhibitors 3-MA, E64, and ConcA (Figure 8). Anti-PEPC antibodies revealed a 65 kDa peptide in the pull-down in all the experiments. In addition, in the presence of 3-MA, an intact 100 kDa peptide was also detected.

Finally, the interaction between ATG8 and PEPC was evaluated using immunoprecipitation following transient expression in *N. benthamiana* (Figure 9). Leaves of *N. benthamiana* were infiltrated with recombinant GFP-ATG8CL. Immunoprecipitation was performed by affinity chromatography with GFP-Trap^®^ A beads (Chromotek), and elution of the proteins from the beads was performed by heating for 5 min at 95 °C. The typical double band, corresponding to monoubiquitinated and no-ubiquitinated PEPC, was revealed by anti-PEPC antibodies in leaf extracts. A 63-kDa anti-PEPC immunoreactive peptide was recovered in the pull-down. Peptide mass fingerprinting and MALDI-MS/MS showed that the peptide was identified as Niben101Scf03628g14021 (NbPPC2) (Appendix A). The higher score was with Niben101Scf03487g00014, which is a fragmented PEPC lacking a N-terminal end. This result suggests an interaction between ATG8 and PEPC, either directly or indirectly via endogenous NBR1.

By using two different bioinformatics tools (iLIR and hfAIM), we found several putative AIM domains in all PEPCs from Arabidopsis and *N. benthamiana* [36,37]; however, with only AtPPC2 and AtPPC4, both methods identified the same motif in the protein (Appendix A). To add another criterion of significance to the results, we used the position-specific scoring matrix (PSSM) index calculated with the iLIR method. This index improves the accuracy of the results with a threshold of 13 for the results to be taken into account as possible AIM domains [37]. Interestingly, only the BTPC isoform in Arabidopsis AtPPC4 followed this criterion, with a PSSM value of 23, suggesting that only this PEPC protein is bound by ATG8; however, the AtPPC4 protein was almost absent in all tissues analyzed in this work [52] and more suitable tissues (i.e., pollen) should be used to certify ATG8 binding. In light of these results, we can conclude that, although it is possible, PTPCs from Arabidopsis and *N. benthamiana* are unlikely to bind ATG8 directly by AIM domains.

## 3. Discussion

Among the most stress-resistant plants, there are many C_4_ and CAM species, particularly for drought and salt stress [53]. In addition, C_3_-type PEPCs have extensively documented roles in response to environmental stresses, such as Al and Cd toxicity [54,55], P starvation [52], and ammonium stress [12]. Work at our laboratory has shown notable results with *Sorghum bicolor*, which has both C_4_ and C_3_ type PEPCs, where the salt and ammonium stresses have increased the amount of PEPC proteins in roots while enhancing the degree of phosphorylation in leaves [12,56]. The mechanisms responsible for these effects are complex, and they include both changes of *PPC* and *PPCK* gene expression and increased stability of PEPC and PEPCk proteins. The mechanisms controlling the degradation of PEPC and the role of PTMs on the stability of the protein are not yet fully understood. This paper has investigated the autophagic degradation of PEPC and the role of monoubiquitination in regulating the stability of PEC proteins in terms of marking towards selective autophagy.

Although there is experimental evidence that shows that both photosynthetic PEPCs and PEPCks can be polyubiquitinated and degraded kDa by the 26S proteasome [57,58], the monoubiquitination of photosynthetic PEPCs has not been reported. This work has focused on C_3_-type PEPCs, investigating whether monoubiquitination has a role in recruiting proteins towards selective autophagy.

Mutations in core ATG genes in Arabidopsis cause defective autophagy and hypersensitivity to various types of nutrient, and also to abiotic and biotic stress factors [59]. These mutants accumulate proteins that are otherwise degraded by autophagy. An increase of the amount of PEPC, and, specifically monoubiquitinated PEPC, was found in the *atg2*, *atg5*, and *atg18a* Arabidopsis lines, which are defective in terms of core ATG proteins. Remarkably, monoubiquitinated PEPC accumulated in the *nbr1* line, which is specifically deficient in selective autophagy. These results suggest that specific autophagy is involved in basal maintenance of PEPC. This is supported by the results of Dr Vierstra’s group. In a maize mutant lacking the core autophagy protein ATG12 (*atg12-1*) [60], C_3_-type PEPC accumulated in old but not in young leaves (Fionn McLoughlin personal communication). Similar results were obtained with the chemical inhibition of autophagic degradation of cargo in cultured tobacco cells. Finally, pull-down experiments demonstrated the interaction between NBR1 and PEPC in Arabidopsis leaves and roots and cultured tobacco cells, as well as between ATG8 and PEPC in *N. benthamiana* leaves.

Monoubiquitinated PEPC was not recovered in the pull-down experiments, as it should have been expected. In addition, intact 100 kDa peptides were rarely found. In most cases, fragmented PEPC peptides were revealed by anti-PEPC antibodies after pull-down. It is possible that monoubiquitinated PEPC is cleaved before its recruitment to the autophagosome. Alternatively, it is also feasible that PEPC is cleaved (for example, by cathepsin proteases) and then monoubiquitinated. Experimental data do not allow distinguishing between these two alternatives. It could be hypothesized that PEPC proteins directed towards selective autophagy are altered forms with a higher sensitivity to proteolytic cleavage. We have recently shown that the interaction of sorghum PEPC with anionic phospholipids changes its conformation and this increases its sensibility to cysteine proteases [61].

Autophagy allows the degradation of damaged elements and the recycling of their components, ensuring good management of resources and preventing the effects of the buildup of dysfunctional elements. This work shows that a fraction of C_3_-type PEPC is cleaved via selective autophagy. In addition, a new function has been attributed to the monoubiquitination of PEPC, marking the protein towards selective autophagy, increasing the knowledge about the mechanisms that control the stability and turnover of the PEPC protein.

## 4. Materials and Methods

### 4.1. Plant Material and Growth Conditions

All *A. thaliana* plants used in this study were from the Columbia (Col-0) background. Seeds of SALK lines *atg18a* and *nbr1* were obtained from the Nottingham Arabidopsis Stock Centre (NASC, London, UK; http://arabidopsis.info). Mutant lines were analyzed, and homozygosity was confirmed by PCR analysis. The *atg2* and *atg5* lines were kindly provided by Dr. Yasin Dagdas (Gregor Mendel Institute of Molecular Plant Biology, Vienna, Austria). Arabidopsis seed surfaces were sterilized with 70% ethanol for 10 min and a 50% HClO solution for 10 min. Finally, the seeds were rinsed 8 to 10 times with sterile water. Seeds were then stratified for 3 d at 4 °C in the dark in a 0.1% agar solution in water to synchronize germination. Seeds were placed in an Araponics^®^ systems on a 0.65% agar and were supplied with half strength Murashige and Skoog (MS) media. Plants were grown with a short day regime, with 8 h in the light (22 °C, 60% relative humidity) and 16 h of dark (18 °C, 70% relative humidity). The light intensity was 140 μmol photons m^−2^ s^−1^ PAR.

*N. benthamiana* seeds were sterilized by washing with 20% bleach (20 min) and several times with distilled water. Plants were grown on a solid substrate in a growth chamber with a day/night cycle of 12 h in the light (25 °C, 60% relative humidity) and 12 h in the dark (20 °C, 70% relative humidity). The light intensity was 350 μmol photons m^−2^ s^−1^ PAR. Plants were grown for 4-5 weeks and then used in experiments for transitory expression after infiltration with *Agrobacterium tumefaciens*.

### 4.2. N. benthamiana Cell Cultures

Surfaced-sterilized seeds were sown on petri dishes containing germination media (50% MS medium, 2% sucrose, 1% agar). After three weeks, roots were excised (0.5 cm) and roots sections were transferred to Petri dishes containing callus induction media (4.4 g L^−1^ MS, 2% glucose, 0.8% agar, 0.5 mg L^−1^ 2,4-dichlorophenoxyacetic acid, 0.05 mg L^−1^ kinetin, 0.5 g L^−1^ MES, pH 5.7). Dishes were sealed with Leukopor tape and kept in continuous white light (20 μmol photons m^−2^ s^−1^ PAR) for 2-3 weeks until calluses were obtained.

Pieces of friable callus (0.3 g) were transferred to a 100 mL flask containing a 20 mL sterile culture medium (4.4 g L^−1^ MS supplemented with Gamborg B5 salts, pH 5.8, 3% sacarose, 0.5 mg L^−1^ α-naphtaleneacetic acid, 0.05 mg L^−1^ kinetin). Cells were kept in an orbital shaker for 10 days under continuous light (100 μmol photons m^−2^ s^−1^ PAR) and then cells were transferred to 80 mL of a fresh culture medium. The cultures were subsequently kept in the dark and subcultured once a week by transferring 25% of the culture to a flask containing 70 mL of a fresh culture medium.

### 4.3. Determination of Enzyme Activity and Protein Quantification

Protein extracts were obtained by grinding 0.2 g fresh weight of leaf or root tissue in a 1 mL extraction buffer containing 0.1 M Tris-HCl, pH 7.5, 20% (*v*/*v*) glycerol, 1 mM EDTA, 10 mM MgCl_2_, a protease inhibitor cocktail (Sigma), 10 mM potassium fluoride, and 14 mM β-mercaptoethanol. The homogenate was centrifuged at 15,000× *g* for 2 min and PEPC activity was quantified in the supernatant.

PEPC activity was measured spectrophotometrically at optimal pH of 8.0 using a NAD-MDH-coupled assay at 2.5 mM PEP [62]. A single enzyme unit (U) is defined as the amount of PEPC that catalyzes the carboxylation of 1 µmol of phospho*enol*pyruvate per minute at a pH of 8 and temperature of 30 °C.

### 4.4. Antibodies

Polyclonal antibodies against C_4_-type PEPC from sorghum leaves (rabbit anti-C_4_ PEPC) were prepared as described in [63] and were used at a ratio of 1:3000. The rabbit anti-ubiquitin antibodies (anti-Ubiquitin) were purchased from Millipore (catalog number 05-944) and used at a ratio of 1:2000. Anti-GST monoclonal antibodies were produced by mice against a recombinant GST protein and obtained from Santa Cruz Biotechnology (product code sc-138). The rat anti-GFP monoclonal antibodies were raised against a recombinant green fluorescent protein and were obtained from Chromotek (product code 3h9-100). Secondary antibodies conjugated to horseradish peroxidase were obtained from: (i) Invitrogen (anti-rabbit, product code 31460); (ii) Novus (anti-mouse, product code NB7539); (iii) CST (anti-rat, product code 7077S). For the chemoluminescent reaction, Supersignal West Pico Chemiluminescent Substrate (Thermo Scientific) and WesternBright Quantum (Advansta) were used, and the Amersham Imagier 600 was used for imaging.

### 4.5. Electrophoresis and Protein Gel Blot Analysis

Samples containing proteins were denatured by boiling (3 min, 90 °C) in the presence of a dissociation buffer (100 mM Tris-HCl, pH 8, 25% [*v*/*v*] glycerol, 1% [*w*/*v*] SDS, 10% [*v*/*v*] β-mercaptoethanol, and 0.05% [*w*/*v*] bromophenol blue). The denatured proteins were separated by SDS-PAGE in a Miniprotean electrophoresis cell (Bio-Rad) and stained with Coomassie Brilliant Blue R-250 or electroblotted onto a nitrocellulose membrane (N-8017 from Sigma) at 10 V (3 mA cm^−2^) for 2 h in a semi-dry blot transfer apparatus (Bio-Rad). Membranes were blocked in Tris-buffered saline (0.02 M Tris-HCl and 0.15 M NaCl, pH 7.5) containing 5% (*w*/*v*) powdered milk, and bands were immunochemically labeled via overnight incubation of the membrane at 4 °C in 20 mL of Tris-buffered saline containing antisera. The Multi Gauge (Fujifilm) software package was used for graphical analysis and the quantification of the immunoblot images.

### 4.6. Recombinant Protein Expression and Purification

Using a Gateway^®^ cloning system, a truncated Sb*NBR1* containing two ubiquitin-associated domains (UBA) and one ATG8 interacting motif (AIM) was cloned into a pGEX-KG vector (kindly provided by Dr. Galván-Ampudia from the University of Amsterdam), then fused to glutathione *S*-transferase (GST). The *Escherichia coli* BL21 strain was used for expression of the GST-NBR1 protein from the previous vector, induced by 1 mM isopropyl β-D-1-thiogalactopyranoside (IPTG) for 6 h. The soluble fraction from the bacterial lysate was incubated with Glutathione Sepharose^®^ 4B (GE Healthcare) to bind GST-NBR1, and the agarose was used for further pull-down experiments.

### 4.7. Pull-Down Experiments

For the pull-down experiments, GFP-Trap^®^_A (Chromotek) and a GST-NBR1 pre-bound agarose (described in Section 4.6) were incubated with protein crude extracts for 2 h (GFP-Trap) or overnight (GST-NBR1) at 4 °C in the presence of a GTEN buffer (25 mM Tris-HCl, pH 7.5, 10% (*v*/*v*) glycerol, 1 mM EDTA, 150 mM NaCl) supplemented with a protease inhibitor cocktail (Sigma), 0.1% (*v*/*v*) IGEPAL^®^ CA-630 (Sigma), and 10 mM DTT. Then, the agarose was washed 5 times with GTEN supplemented with 0.1% (*v*/*v*) IGEPAL^®^ CA-630 and the proteins were eluted by denaturation and heating.

### 4.8. Protein Databases Searches, Alignment, and Phylogenetic Analysis

Searches for *A. thaliana* PEPCs (AtPEPC1-4) were carried out with the NCBI database. The homolog sequences for *N. benthamiana* and homology index values were obtained using the BlastP server available for the draft sequence *N. benthamiana* genome database via the SOL Genomics Network from the Boyce Thompson Institute for Plant Research (https://solgenomics.net/organism/Nicotiana_benthamiana/genome) [8]. Partial sequences were discarded. The selected PEPC protein sequences for *N. benthamiana* were named in terms of reference to their homology as follows: NbPPC1 (Niben101Scf25430g00015), NbPPC2 (Niben101Scf03628g14021), NbPPC3.1 (Niben101Scf00031g00003), NbPPC3.2 (Niben101Scf04036g04008), NbPPC3.3 (Niben101Scf03439g03004), and NbPPC4 (Niben101Scf00312g03005).

Alignments of the amino acid PEPCs sequences from *N. benthamiana* and *A. thaliana* were performed automatically using MUSCLE [64], implemented in MEGA version 10.0.5 [65]. Bayesian inference (BI) and maximum likelihood (ML) analyses were performed using MrBayes 3.2 [66] through the CIPRES science gateway [67] and RAxML 7.2.6 [68] through the T-Rex web server [69], respectively. Bayesian inference analysis was carried out as in [70], using the amino acid substitution model that best fit our data as predicted by ProtTest 3 [71], based on the highest Akaike’s Information Criterion weights (AICw) [72]. In the case of the ML analysis, a PROTCAT model and Dayhoff matrix substitution model [73] were used. Bootstrap support (BS) values were calculated using a rapid bootstrapping algorithm with 10,000 rapid bootstrapping searches and 2000 ML searches to estimate the best tree. Bootstrap values under 70% were considered non-significant. Trees were finally edited using FigTree version 1.4.0 (http://tree.bio.ed.ac.uk/software/figtree/). Moreover, a pairwise evolutionary distance matrix was calculated with a gamma distribution of among-site rate variation as implemented in MEGA version 10.0.5 [64].

### 4.9. Protein Digestion and Mass Spectrometry Identification

Protein identification was accomplished by matrix-assisted laser desorption ionization time-of-flight mass spectrometry (MALDI-TOF-MS). Protein samples were analyzed by SDS-PAGE and stained with Coomassie Brilliant Blue R-250. Bands were excised with a cutting edge and de-stained afterwards using 25 mM ammonium bicarbonate (AB) and acetonitrile. Disulfide bonds were reduced by embedding bands in 200 µL of 10 mM DTT and 50 mM AB, which was maintained for 60 min at 56 °C. After reduction, the samples were incubated in 200 µL of 30 mM IAA and 50 mM AB for 30 min at room temperature in a dark environment in order to block Cys-SH groups.

Proteins were digested overnight at 37 °C using trypsin bovine (Sequencing Grade Modified Trypsin, Promega) at a ratio of 1:10 for the enzyme and substrate, respectively. After digestion, acetonitrile and a trifluoroacetic acid 0.2% solution were added to the bands for peptide extraction. OMIX C18 tips (Agilent Technologies) were used for concentrating and desalting peptide extracts. Next, 0.5 µL of each sample was spotted onto a MALDI sample plate. After solvent evaporation, 0.5 µL alpha-cyano-hydroxycinnamic acid saturated solutions were spotted over the sample spots and air-dried.

MALDI fingerprint spectra were obtained using a MALDI-TOF Ultraflextreme (Bruker) system in the positive ion reflectron mode, summing 5000 shots for every spectrum. The FlexAnalysis software package was used to calibrate, select, and filter mass peaks. MS data were analyzed using the MASCOT search engine with the Biotools™ proteomic software package (Bruker) using static carbamidomethylation (C) and dynamic oxidation (M) modifications. Data were searched against the Uniprot *A. thaliana* and *N. benthamiana* protein database and against PEPC amino acid sequences. Peptides of interest were further analyzed by tandem mass spectrometry (MALDI TOF-MS/MS) in order to assess their sequences.

## Figures and Tables

**Figure 1 plants-10-00012-f001:**
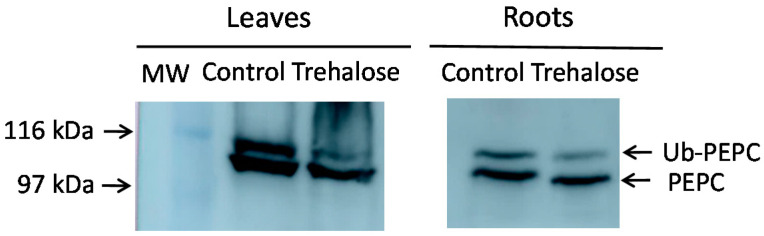
Trehalose decreases the monoubiquitination of Arabidopsis phospho*enol*pyruvate carboxylase (PEPC). Arabidopsis plants were hydroponically grown and, when indicated, 1% (29 mM) trehalose was supplied to the culture medium for 48 h. The leaves and roots from six-week-old plants were pooled (30 plants). Protein aliquots corresponding to 10 mU of PEPC were analyzed by SDS-PAGE and immunoblotted with anti-PEPC antibodies. The arrows show monoubiquitinated (Ub-PEPC) and non-monoubiquitinated PEPC. MW, molecular mass markers. The two images are of the same membrane, although several lanes are omitted for clarity.

**Figure 2 plants-10-00012-f002:**
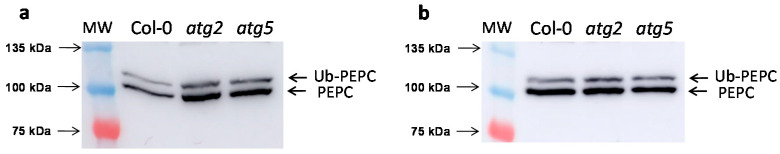
Arabidopsis *atg2* and *atg5* autophagy-defective SALK lines accumulate PEPC. Leaves from six-week-old Arabidopsis Col-0, *atg2*, and *atg5* SALK lines were pooled (30 plants for sample). Protein aliquots from crude extracts were analyzed by SDS-PAGE and immunoblotted with anti-PEPC antibodies. (**a**) Results for 5 mU of PEPC from leaf extracts. (**b**) Results for 50 µg of proteins from leaf extracts.

**Figure 3 plants-10-00012-f003:**
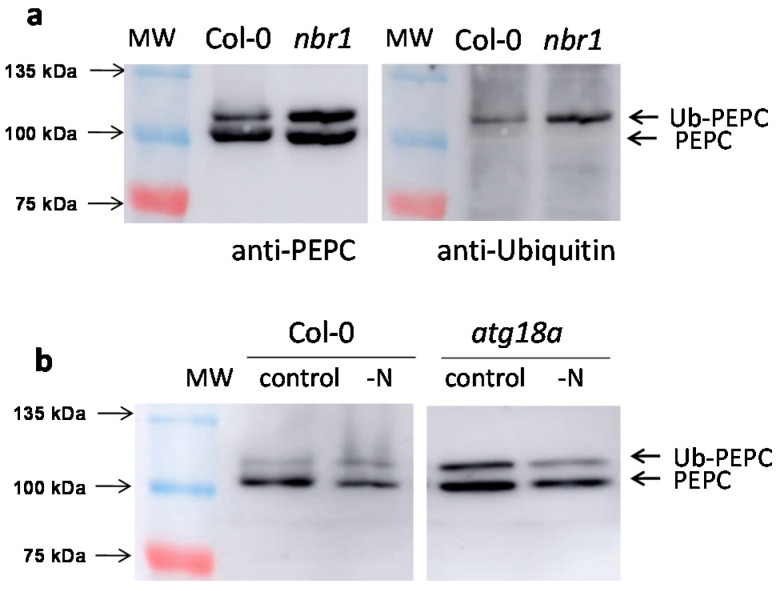
The amount of monoubiquitinated PEPC increases in the *nbr1* and *atg18a* SALK lines. Arabidopsis Col-0, *nbr1*, and *atg18a* SALK lines were hydroponically grown for six weeks. Leaves from 30 plants were pooled and protein aliquots corresponding to 10 mU of PEPC were analyzed by SDS-PAGE. (**a**) Leaf extracts from Col-0 and *nbr1*. Immunoblots with anti-PEPC and anti-ubiquitin antibodies of the same membrane. (**b**) Leaf extracts from Col-0 and *atg18a*. When indicated, N was suppressed from the culture medium for 1 week. Several lanes are omitted for clarity.

**Figure 4 plants-10-00012-f004:**
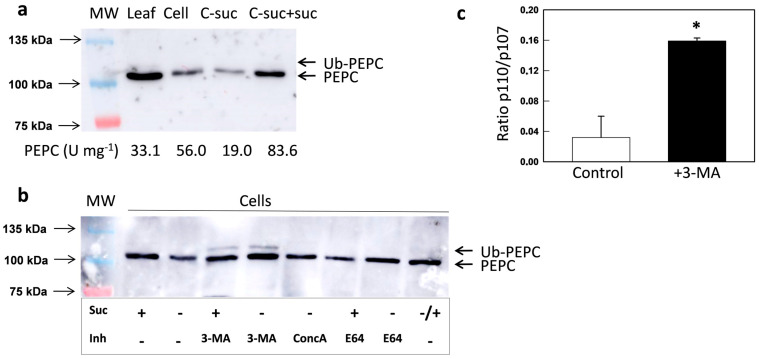
Effect of sucrose starvation and inhibitors on the PEPC of *N. benthamiana* culture cells. Crude extracts were prepared with leaves or cultured cells. Protein aliquots corresponding to 30 µg were analyzed by SDS-PAGE and immunoblotted with anti-PEPC antibodies. (**a**) When indicated, sucrose was suppressed from the culture medium of cells for 5 d (C-suc) or suppressed for 5 d and then resupplied for 2 d (C-suc+suc). The numbers under the immunoblots show the PEPC activities of extracts. (**b**) Cells were cultured with a control medium for 5 days. When indicated, sucrose was suppressed (5 d) or suppressed for 5 d and then resupplied for 2 d (−/+). 3-MA was added at 2.5 mM (5 d), along with ConcA at 1 µM (16 h) and E64 at 10 µM (16 h). (**c**) The ratio of p110/p107 (Ub-PEPC/PEPC) of the signal with or without 3-MA was calculated. * *p* < 0.05 versus Col-0 (*t*-test, *n* = 3).

**Figure 5 plants-10-00012-f005:**
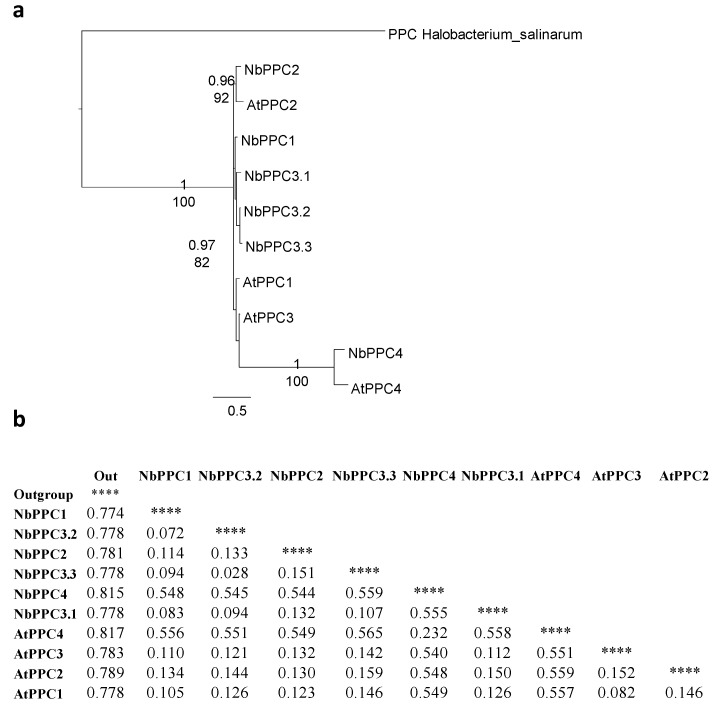
Phylogenetic analysis of *N. benthamiana* PEPCs. (**a**) Phylogeny of phospho*enol*pyruvate carboxylase (PPC) proteins and their isoforms in *N. benthamiana* and *A. thaliana*. For protein identification, “Nb” indicates *N. benthamiana* proteins and “At” indicates *A. thaliana* proteins. As outgroup was used a PEPC from *Halobacterium salinarum* (HsPPC). Bootstrap analysis was carried out with 100 replicates. Number at the branches correspond to the bootstrap frequencies for each branch. (**b**) Evolutionary divergence matrix estimated from PEPCs proteins sequences. The rate variation among sites was modeled with a gamma distribution (shape parameter = 1). **** means “not applicable”.

**Figure 6 plants-10-00012-f006:**
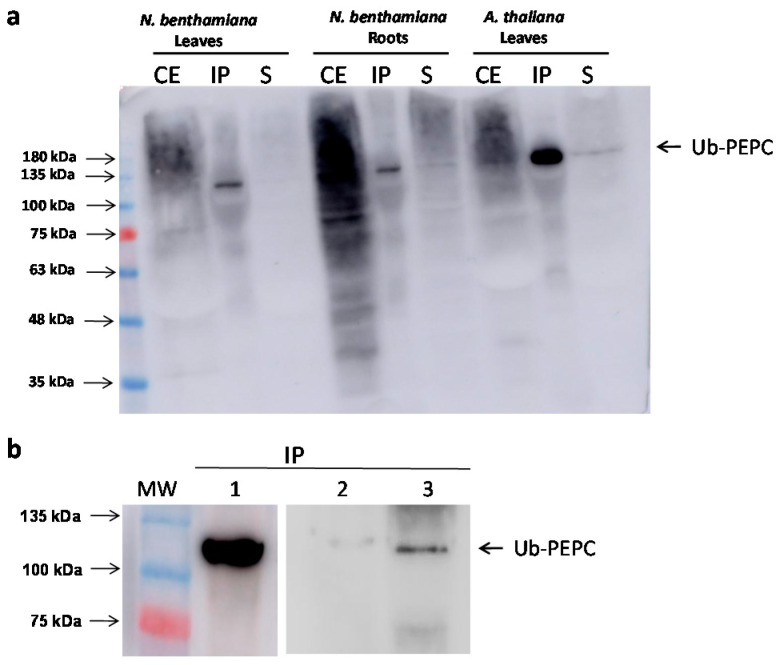
Monoubiquitinated PEPC is revealed by anti-Ubiquitin antibodies in immunoprecipitates. Crude extracts (CE) were obtained from different plant tissues (0.5 g/mL extraction buffer for leaves, and 1 g/mL extraction buffer for roots and cells) and PEPC contained in 3 mL of crude extracts was immunoprecipitated with anti-PEPC antibodies. Finally, immunoprecipitates were analyzed by immunoblotting and detected with anti-Ubiquitin antibodies. (**a**) PEPC immunoprecipitation from *Nicothiana benthamiana* leaves and roots and *A. thaliana* leaves. CE, 30 µg protein from crude extract; IP, immunoprecipitate; S, supernatant. (**b**) PEPC immunoprecipitated from extracts of *A. thaliana* leaves (lane 1), *N. benthamiana* cells (lane 2), and 3-MA-treated *N. benthamiana* cells (lane 3).

**Figure 7 plants-10-00012-f007:**
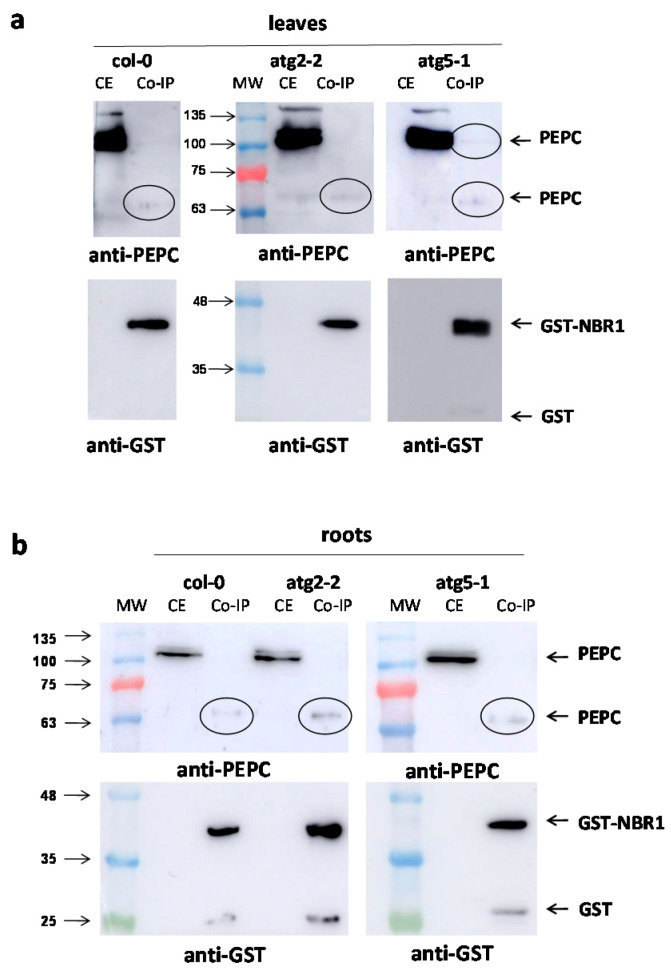
Co-immunoprecipitation of PEPC with GST-NBR1 in Arabidopsis lines. Crude extracts from leaf (**a**) and root (**b**) tissues of Col-0, *atg2*, and *atg5* Arabidopsis (six weeks) were incubated overnight at 4 °C with purified GST-NBR1 pre-bound to Gluthatione Sepharose^®^ 4B (GE Healthcare). Then, extracts were centrifuged at 1000× *g* for 2 min at 4 °C, washed five times, and co-precipitated proteins were eluted by heating at 95 °C for 5 min with a dissociation buffer. Afterwards, 50 μg of proteins from crude extracts (CE) and corresponding amounts of co-precipitated proteins (Co-IP) were analyzed by SDS-PAGE and immunoblotted with anti-PEPC and anti-GST antibodies.

**Figure 8 plants-10-00012-f008:**
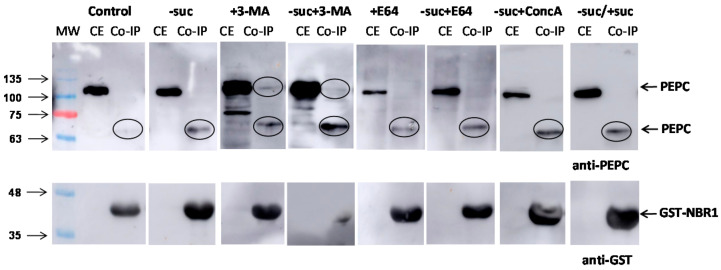
Co-immunoprecipitation of PEPC with GST-NBR1 in *N. benthamiana* cells. Crude extracts were prepared from five-day-old *N. benthamiana* cell cultures and aliquots containing 50 µg of protein were used in the pull-down experiments. When indicated, cells were kept without sucrose (-suc), or 5 d without sucrose and then 2 d with sucrose (−suc/+suc). 3-MA was added at 2.5 mM (5 d), along with ConcA at 1 µM (16 h), and E64 at 10 µM (16 h). The other conditions are the same as in Figure 6. CE, crude extracts; Co-IP, co-precipitated proteins.

**Figure 9 plants-10-00012-f009:**
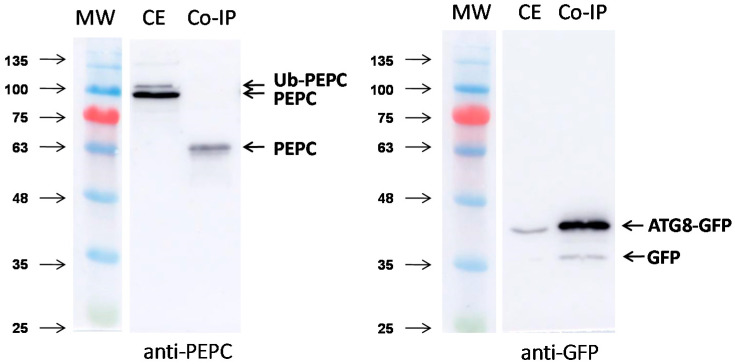
Co-immunoprecipitation of PEPC with GFP-ATG8CL in *N. benthamiana*. A transformed *Agrobacterium tumefaciens* strain GV3101 line carrying potato ATG8CL fused to green fluorescent protein (GFP) has infiltrated *N. benthamiana.* At 60 h after agroinfiltration, proteins were extracted from infiltrated leaves using a 2 mL GTEN buffer per g of tissue. Crude extracts were incubated with GFP-Trap^®^_A (Chromotek) for 2 h at 4 °C. Then, extracts were centrifuged at 1000 *g* for 2 min at 4 °C, washed five times, and co-precipitated proteins were eluted by heating at 95 °C for 5 min with a dissociation buffer. Afterwards, 15 μL of the crude extracts (CE) and co-precipitated proteins (Co-IP) were immunoblotted with the anti-PEPC (left) and anti-GFP (right) antibodies.

**Table 1 plants-10-00012-t001:** The amounts of monoubiquitinated PEPC increased in autophagy-defective Arabidopsis lines.

	Ratio of p110/p107	
	Leaves	Roots
Col-0	0.34 ± 0.04	0.11 ± 0.02
*atg2*	0.53 ± 0.04	0.18 ± 0.02
*atg5*	0.48 ± 0.03	0.16 ± 0.02
*nbr1*	0.58 ± 0.16	0.38 ± 0.19
*atg18a*	0.50 ± 0.07	0.19 ± 0.03

Leaves or roots from six-week-old Arabidopsis Col-0, atg2, atg5, nbr1, and atg18a SALK lines were pooled (30 plants for sample). Proteins aliquots from crude extracts from leaf or roots were analyzed by SDS-PAGE and immunoblotted with anti-PEPC antibodies. The ratios (Ub-PEPC/PEPC) of the signals were calculated. The table shows quantitative data (mean ± SE, *n* = 5) for the ubiquitination.

## Data Availability

The data presented in this study are available in article and supplementary material.

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
