# Peer review of "Genetic and Pharmacological Inhibition of Autophagy Increases the Monoubiquitination of Non-Photosynthetic Phosphoenolpyruvate Carboxylase"

_plants, 2020, doi:10.3390/plants10010012_

Round 1

Reviewer 1 Report

This study investigated the modification of a key enzyme PEPC involving in Carbon and Nitrogen metabolism. They showed that PEPC was monoubiquitinated and this forms was degraded by autophagy. This form and the total protein level of PEPC were accumulated in mutants that are genetically deficient in autophagy pathway. Treatment of chemical inhibitor of autophagy also increased the monoubiquitinated form of PEPC. And the authors presented data supporting the interaction of PEPC with NBR1, ATG8 in Arabidopsis or N. Benthamiana. The manuscript was very well written, and the data were clearly presented. I do not have any concerns that are needed to be addressed.

Author Response

RESPONSE: The authors thank the reviewer for revising the paper

Reviewer 2 Report

In this manuscript, the authors suggested that monoubiquitinated PEPC was targeted to selective autophagy for cleavage. Although this work is potentially interesting, the authors need to provide more convincing evidence and thorough interpretation of their data to improve the manuscript. The descriptions of results are sometimes too concise to understand the author’s work. Furthermore, some of the experimental results do not fully support the author’s conclusions. Especially, it is hard to believe that monoubiquitinated PEPC is cleaved by autophagy, as autophagic degradation does not usually yield a cleaved peptide. Rather, it is possible that monoubiquitinated PEPC is cleaved by some proteases, and then this cleaved peptide is targeted to selective autophagy. The following points need to be addressed.

Specific comments

  1. For most Figures, the authors need to provide a loading control of the immunoblot. The loading control should be the one that is widely used in plant biology. The use of a loading control is for more accurate comparison between experimental groups and controls.
  2. In Figure 1~4, the authors described that autophagy inhibition via genetic alteration of autophagy-related proteins or chemical reagents led to increase of PEPC and monoubiquitinated PEPC levels. However, as the basal level of autophagy activation or clearance seems to be low in their model plants, they need to check the levels of PEPC and monoubiquitinated PEPC after autophagy activation to further strengthen their results. In particular, the critical results supporting their conclusions will be drawn by comparing PEPC levels among the control group, the autophagy-activated group, and the group treated with autophagy activator and inhibitor simultaneously.
  3. In Figure 4b lane 6 (+, E64), PEPC level seems to be decreased compared to the control despite the treatment of autophagy inhibitor E64, which was opposite to our expectation. The authors should explain this result.
  4. In co-IP results (Figures 7~9), full-length PEPC or monoubiquitinated PEPC did not seem to interact with autophagy-related proteins. As cleaved form of PEPC (63 or 65 kDa, please clarify) interacted with NRB1 or ATG8, it is possible that, for autophagic degradation of PEPC, the cleavage is required and is more important than monoubiquitination. The authors need to discuss about this possibility. Is there any likelihood that monoubiquitination is a signal or prerequisite for the cleavage of PEPC? What is the evidence that the cleavage of PEPC is the result of autophagy?

Minor comment

  1. Replace the word 'autofagosomas' in line 194.
  2. It is confusing whether PEPC is p100 or p107. If monoubiquitinated PEPC is p110, this should be the sum of PEPC’s MW and ubiquitin’s MW (~7 kDa).

Author Response

REVIEWER Q1: In this manuscript, the authors suggested that monoubiquitinated PEPC was targeted to selective autophagy for cleavage. Although this work is potentially interesting, the authors need to provide more convincing evidence and thorough interpretation of their data to improve the manuscript. The descriptions of results are sometimes too concise to understand the author’s work. Furthermore, some of the experimental results do not fully support the author’s conclusions.

Especially, it is hard to believe that monoubiquitinated PEPC is cleaved by autophagy, as autophagic degradation does not usually yield a cleaved peptide. Rather, it is possible that monoubiquitinated PEPC is cleaved by some proteases, and then this cleaved peptide is targeted to selective autophagy.

RESPONSE 1: We agree with the reviewer that is not likely that fragmented PEPC peptides are produced by autophagic proteolysis. This point is further discussed in the response to specific comments (RESPONSE 5).

REVIEWER: The following points need to be addressed.

SPECIFIC COMMENTS

REVIEWER Q2: For most Figures, the authors need to provide a loading control of the immunoblot. The loading control should be the one that is widely used in plant biology. The use of a loading control is for more accurate comparison between experimental groups and controls.

RESPONSE 2: When we initiated the study of autophagy-deficient Arabidopsis lines, we discussed which would be the best loading control for immunoblots. In these Arabidopsis lines, we would expect alterations of many proteins. As we wanted to evaluate the fate of PEPC among the whole proteome, we chose to quantify proteins (Bradford) and to use the amount of protein as loading control. This amount is detailed in Figure legends. In some experiments (Figure 1, Figure 2a, Figure 3), in order to detect inactive protein PEPC, we used specific PEPC activity as loading control. This represents the amount of protein needed to get a predetermined PEPC activity; the amount would be greater if inactive PEPC accumulates.

REVIEWER Q3: In Figure 1~4, the authors described that autophagy inhibition via genetic alteration of autophagy-related proteins or chemical reagents led to increase of PEPC and monoubiquitinated PEPC levels. However, as the basal level of autophagy activation or clearance seems to be low in their model plants, they need to check the levels of PEPC and monoubiquitinated PEPC after autophagy activation to further strengthen their results. In particular, the critical results supporting their conclusions will be drawn by comparing PEPC levels among the control group, the autophagy-activated group, and the group treated with autophagy activator and inhibitor simultaneously.

RESPONSE 3: We agree with the reviewer with respect to the experimental approach. In fact, we have subjected the Arabidopsis lines to different conditions that are known to trigger autophagy: oxidative stress caused by methyl viologen treatment, salinity and N-starvation. In all these conditions, monoubiquitinated PEPC did not accumulated. We interpreted that other mechanisms, such proteases or the proteasome, which are activated by stress, should be responsible for PEPC degradation. These results suggest that specific autophagy is involved in basal maintenance of PEPC. This is supported by results of Dr Vierstra´s group. In a maize mutant lacking the core autophagy protein ATG12 (atg12-1), C3-type PEPC accumulated in old but not in young leaves (McLoughlin personal communication).

            To clarify this point, new experimental data has been included in Figure 3b, and discussed in the text of the manuscript (lines 174-179 and lines 355-358 ).

REVIEWER Q4: In Figure 4b lane 6 (+, E64), PEPC level seems to be decreased compared to the control despite the treatment of autophagy inhibitor E64, which was opposite to our expectation. The authors should explain this result.

RESPONSE 4: E64 treatment in non-stressed cells (+, E64; Figure 4b lane 6) decreased total level of PEPC protein compared to the control (+, -; Figure 4b lane 1). Both samples are control samples in which autophagy is not triggered, as there is no nutrient deficiency. E64 is a cysteine protease inhibitor, which has been reported to suppress autophagy at its latest stage. Nevertheless, E64 is not specific for autophagy inhibition, as it has been described to alter other cell pathways as cell death and embryogenesis induction (Bárány et al., 2018. J Exp Bot 69, 1387-1402). We hypothesize that in absence of stress, the cysteine protease inhibitor E64 interferes with other cellular pathways which leads into reduced total PEPC protein. In spite of the previous result, when compared between both E64 samples (Figure 4b lane 6 [+, E64], Figure 4b lane 7 [-, E64], an increase in total PEPC levels is shown, as a result of the inhibition of  autophagy by E64 in carbon-stressed cells.

REVIEWER Q5: In co-IP results (Figures 7~9), full-length PEPC or monoubiquitinated PEPC did not seem to interact with autophagy-related proteins. As cleaved form of PEPC (63 or 65 kDa, please clarify) interacted with NRB1 or ATG8, it is possible that, for autophagic degradation of PEPC, the cleavage is required and is more important than monoubiquitination. The authors need to discuss about this possibility. Is there any likelihood that monoubiquitination is a signal or prerequisite for the cleavage of PEPC? What is the evidence that the cleavage of PEPC is the result of autophagy?

RESPONSE 5: We agree with the reviewer in his/her statements, and we do not believe that the cleavage of PEPC is the result of autophagy. There are other two more feasible possibilities: i) monoubiquitinated PEPC is cleaved, and afterwards recluted into the autophagosome; ii) PEPC is cleaved (for example by cathepsin proteases) and then monoubiquitinated. Experimental data do not allow us to distinguish between these two alternatives. To further clarify this item, the following sentence has been included in the manuscript:

Lines 365-368: It is possible that monoubiquitinated PEPC is cleaved before its recruitment to the autophagosome. Alternatively, it is also feasible that PEPC is cleaved (for example, by cathepsin proteases) and then monoubiquitinated. Experimental data do not allow distinguishing between these two alternatives. 

MINOR COMMENT

REVIEWER Q6: Replace the word 'autofagosomas' in line 207.

RESPONSE 6: replaced by “autophagosomes”

REVIEWER Q7: It is confusing whether PEPC is p100 or p107. If monoubiquitinated PEPC is p110, this should be the sum of PEPC’s MW and ubiquitin’s MW (~7 kDa).

RESPONSE 7: P107 and p110 refer to the relative mobility of the protein in SDS-PAGE (movement of a polypeptide through a gel relative to other protein bands in the gel) and do not match exactly with PEPC molecular weight or PEPC+Ubiquitin molecular weight. P107 and p110 have extensively used in our publications and Dr Plaxton´s group publications to define de-ubiquitinated and ubiquinated PEPC, respectively.

The authors thank the reviewer his/her help to improve the paper

Reviewer 3 Report

Comments

Fig. 5:

1. Unify the notations in a) and b)

2. w a) MbPPC3L1?

3. w a) PPC4-L4?

4. in b) replace commas in numbers with dots

L. 486-488: S.G.M. ?

L. 487:  L.H.H.?

Author Response

RESPONSE: Figure 5 has been changed as requested (the notations in a) and b) have been unified, Mb changed to Nb, and commas replaced by dots)

Lines 512-514: “S.G.M.“ has been replaced by S.G.

Line 513: “L.H.H.“ has been replaced by L.H.

The authors thank the reviewer his/her help to improve the paper

Reviewer 4 Report

The manuscript "Genetic and pharmacological inhibition of autophagy increases monoubiquitination of non-photosynthetic phosphoenolpyruvate carboxylase  submitted to Plants has been reviewed. It investigates the degradation of PEPC via selective autophagy, and the role of
monoubiquitination of the enzyme in this process.

The experiment setup was in logic. Data was collected and analyzed properly, and results support the conclusions. The author also explained why some of the results were not as expected. I would suggest the manuscript needs a final proofreading to correct errors and preferably a English polishing.

Author Response

RESPONSE: We have revised carefully the manuscript following the indications of the reviewer

  • The manuscript has been processed by the English editing service of MDPI
  • The manuscript has been modified as indicated by the reviewer in the annotated file as follows

Line 32: “aminoacids” changed to amino acid

Line 35: “CAM” changed to “Crassulacean Acid Metabolism (CAM)”

Line 38: “PEPC gene family include” changed to “The PEPC gene family includes

Line 47: “monoubiquitinatión” changed to monoubiquitination

Line 97: The response to the question “Budding, fission or both?” is “both” (Shpilka, T., Weidberg, H., Pietrokovski, S.et al. Atg8: an autophagy-related ubiquitin-like protein family. Genome Biol 12, 226 (2011). https://doi.org/10.1186/gb-2011-12-7-226)

Lines 368-370: There are many reports establishing that a key function of autophagy is the clearance of altered proteins. In addition, altered proteins can be more susceptible towards proteases. With respect to PEPC, we are at present studying cathepsin proteases that co-purify with and proteolize PEPC (Gandullo et al. 2019). The sensibility of PEPC to these proteases depends on several factors. For example, in PEPC solutions that have been extending stored, and that presumably contain altered forms of PEPC, the sensibility to proteases is highly enhanced.

Line 386: “during” changed to “for”

Line 463: “µ” has been changed to µl

The authors thank the reviewer his/her effort to improve the paper

Reviewer 5 Report

The current version of the paper entitled: "Genetic and pharmacological inhibition of autophagy 3 increases monoubiquitination of non-photosynthetic 4 phosphoenolpyruvate carboxylase" is well presented and structured and all the experiments have been carried out properly and the data analyzed and interpreted as expected.

Considering these premises, I recommend the paper for publication after minor revisions.

I suggest always use the abbreviation of Nicotiana benthamiana and Arabidopsis thaliana throughout the manuscript.

Lines 214: use italic font for N. benthamiana

Author Response

  1. benthamiana and A. thaliana have been used for Nicotiana benthamiana and Arabidopsis thaliana throughout the manuscript

Line 214: Italics have been used for N. benthamiana

The authors thank the reviewer his/her help to improve the paper